# 3D-ViTac: Learning Fine-Grained Manipulation with Visuo-Tactile Sensing

**Binghao Huang**[1]  **Yixuan Wang**[1]  **Xinyi Yang**[2]  **Yiyue Luo**[3]  **Yunzhu Li**[1]

Columbia Univerisity[1]   University of Illinois Urbana-Champaign[2]
University of Washington[3]

**Abstract:** Tactile and visual perception are both crucial for humans to perform fine-grained interactions with their environment. Developing similar multi-modal sensing capabilities for robots can significantly enhance and expand their manipulation skills. This paper introduces **3D-ViTac**, a multi-modal sensing and learning system designed for dexterous bimanual manipulation. Our system features tactile sensors equipped with dense sensing units, each covering an area of $3mm^2$. These sensors are low-cost and flexible, providing detailed and extensive coverage of physical contacts, effectively complementing visual information. To integrate tactile and visual data, we fuse them into a unified 3D representation space that preserves their 3D structures and spatial relationships. The multi-modal representation can then be coupled with diffusion policies for imitation learning. Through concrete hardware experiments, we demonstrate that even low-cost robots can perform precise manipulations and significantly outperform vision-only policies, particularly in safe interactions with fragile items and executing long-horizon tasks involving in-hand manipulation. Our project page is available at https://binghao-huang.github.io/3D-ViTac/.

**Keywords:** Contact-Rich Manipulation, Multi-Modal Perception, Tactile Sensing, Imitation Learning

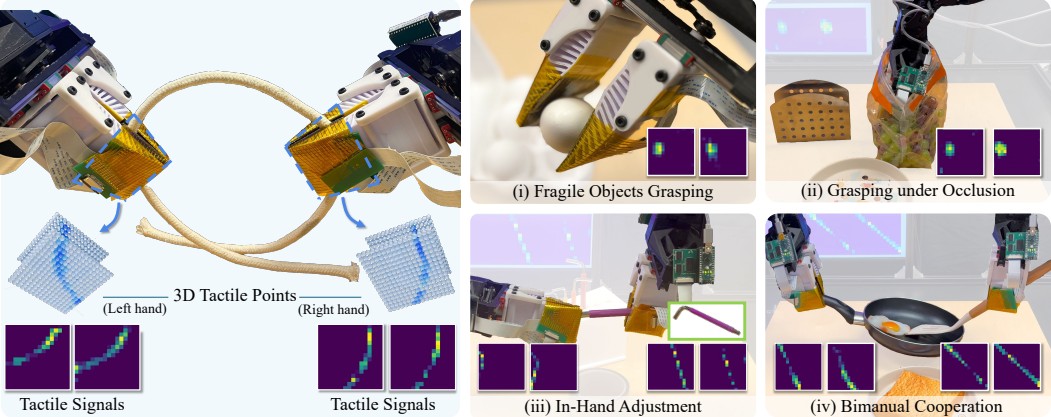

Figure 1: We propose **3D-ViTac**, a multi-modal sensing and learning system for dexterous bimanual manipulation. This system features flexible, scalable, low-cost tactile sensors, each finger equipped with a $16 \times 16$ sensor array. To demonstrate the capabilities of our system in performing precise manipulations, we showcase four tasks that utilize the force-related and in-hand position information provided by the tactile sensors.

## 1   Introduction

Humans heavily rely on both visual and tactile sensing to perform everyday manipulation tasks. Consider grasping an egg or a grape: we start by visually locating the object, and then extract more information from tactile interactions with the object to determine the appropriate amount of force to apply. When eating with a spoon, our eyes estimate the global position and geometric information of the spoon, while our sense of touch provides detailed contact information during interactions with the

8th Conference on Robot Learning (CoRL 2024), Munich, Germany.

food. Vision and touch complement each other, enhancing our interaction with the environment and significantly increasing our flexibility and robustness, especially in tasks involving large occlusions and in-hand manipulation.

However, building such a multi-modal platform for robots brings two major challenges: (i) *Tactile Hardware and System.* Due to requirements for minimal range and viewing directions, many existing optical tactile sensors can be either overly bulky or overly rigid for fine-grained manipulation or tasks that require compliant interactions [1–3]. Many commercial tactile sensors can also be expensive due to customized manufacturing and electronic components [4, 5], while some low-cost tactile sensors are usually too sparse to convey effective information [2, 6–8]. (ii) *Distinct Nature of Tactile and Visual Modalities.* Tactile signals are typically local and physical, while visual data are more global and semantic. This disparity poses significant challenges for models to process and interpret the data effectively. Successfully fusing these distinct modalities requires careful design of the tactile sensor and the learning algorithm.

To address these challenges, we present **3D-ViTac**, a novel multi-modal sensing and learning system for contact-rich manipulation tasks. (i) For the tactile sensing hardware system, rather than using existing optical tactile sensors [2, 5, 9, 10], we propose an alternative dense, flexible tactile sensor array that covers a larger area of the robot end-effector. Our tactile sensors, inspired by the STAG glove [11], are cost-effective, flexible, and capable of producing stable continuous signals during manipulation. As illustrated in Fig. 1, the sensor array has a resolution of $16\times16$ on each soft finger, totaling 1024 tactile sensing units across our bimanual tactile sensing system. This dense, continuous tactile sensor array provides effective feedback, including the presence of contact, the amount of applied normal force, and local contact patterns. (ii) On the algorithm side, given the multi-modal sensory information, instead of separately inputting visual and tactile data into the policy [12], we propose a unified 3D visuo-tactile representation that fuses these two modalities for imitation learning. This representation integrates 3D visual points and 3D tactile points (calculated using robot kinematics) into a unified 3D space, which explicitly accounts for the 3D structure and spatial relationship between vision and touch. This approach enables effective imitation learning through diffusion policy [13], allowing the system to react to nuanced force changes and overcome significant visual occlusions.

We conducted comprehensive evaluations on four challenging real-world tasks (e.g., manipulating fragile objects like eggs and fruit, and in-hand manipulation of tools and utensils, as shown in Fig. 1). The results demonstrate that our 3D visuo-tactile representation significantly enhances the performance of contact-rich manipulations by providing more detailed contact states and local geometry or position information. We observed that tactile information is especially critical when there is heavy visual occlusion. We also conducted detailed ablation studies regarding the sensing characteristics, comparing performance across different tactile resolutions, and showed the importance of continuous tactile reading. Additionally, the inclusion of tactile feedback during the data collection process enables the operator to gather higher-quality data, making the final policy more robust.

## 2 Related Work

**Bimanual Manipulation.** Dual-arm robotic setups present a wealth of opportunities for broad applications [13–17]. Traditionally, approaches to bimanual manipulation were based on a classical model-based control perspective using known environmental dynamics [18–25]. However, these methods depend on ground truth environment models that are not only time-consuming to construct but also typically require full-state estimation, which is often difficult to obtain, particularly for objects with complex physical properties, such as deformable items. In recent years, many researchers in the robotics community have increasingly shifted their focus to learning-based methods, such as reinforcement learning [15, 26–32], and imitation learning [16, 17, 33–41]. However, most bimanual manipulation methods still primarily rely on visual inputs [13, 16, 42–46], limiting the robot's ability to achieve human-level flexibility and dexterity due to the sensing gap between humans and robots. To overcome these limitations, recent works [47, 48] employ RGB images from optical tactile sensors. However, due to the minimal range of the cameras in these sensors, the robot fingers are typically very bulky and overly rigid, limiting their effectiveness in more complicated dexterous tasks. In

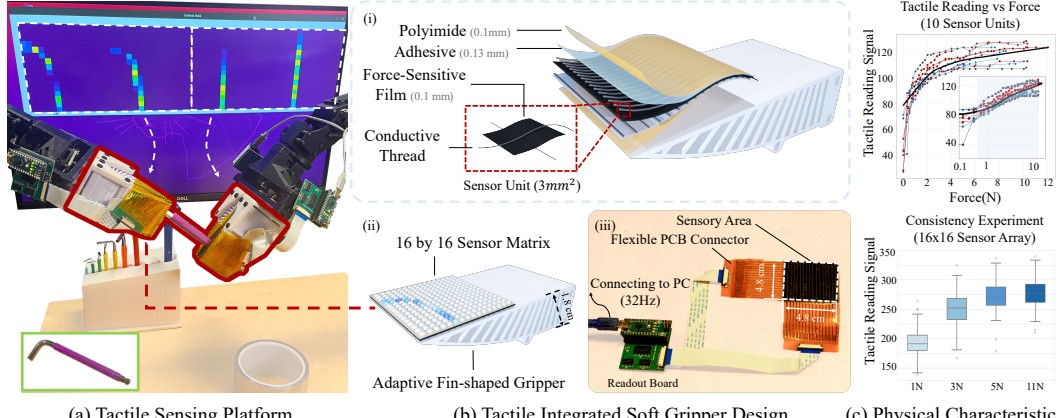

| (a) Tactile Sensing Platform | (b) Tactile Integrated Soft Gripper Design | (c) Physical Characteristic |

Figure 2: **Our Tactile Sensing Platform. Part (a)** shows our bimanual tactile integrated system setup. We deploy four tactile sensor pads (two for each hand) on the soft grippers. The tactile readings are displayed on the back screen. **Part (b)** describes the design of our tactile-integrated soft gripper. Each sensor comprises 256 sensing units, with their locations on the gripper shown in (ii). We have also designed a readout board to collect tactile signals and forward them to the host computer. **Part (c)** shows the physical characteristics and sensing consistency of our tactile sensors (details in Sec. 5).

contrast, the tactile sensor introduced in this work is based on piezoresistive materials, allowing for larger-scale, compliant coverage of flexible thin fingers.

**Visuo-Tactile Manipulation.** Tactile information plays a crucial biological role [49], and the integration of vision and touch is fundamental for humans to successfully interact with their environment [50]. Vision provides a broad perspective of the environment but often lacks detailed contact information and suffers from visual occlusion, which can be effectively complemented by tactile sensing [51–62]. Integrating visual and tactile information is also crucial for robotic manipulation. Lin *et al.* [14] propose a visuo-tactile policy that leverages human demonstrations within a bimanual system. However, their tactile sensor is low-resolution, and their approach lacks an explicit account of the spatial relationship between vision and touch. To address this limitation, Yuan *et al.* [27] proposed Robot Synesthesia, which combines visual and tactile data as a single input to the policy network. However, this approach only considers low-resolution binary tactile signals, which are limited in visuo-tactile sensing capacity. In contrast, our method employs a dense continuous tactile sensing system that delivers comprehensive information about the contact area. We also provide explicit accounting of the scene structure by integrating both modalities into a unified 3D representation space, leading to a more effective policy learning process.

## 3 Visuo-Tactile Manipulation System

### 3.1 Sensor and Gripper Design

**Flexible Tactile Sensors.** Our sensor pads consist of resistive sensing matrices that convert mechanical pressure into electrical signals. Designed with a total thickness of less than $1mm$, the tactile sensor pads can be easily integrated onto various robotic manipulators, including the surfaces of robot arms. In this paper, we install the tactile sensors on a soft and adaptable fin-shaped gripper, as shown in Fig. 2(b). These flexible sensor pads bend with the soft gripper and continue to provide effective signal transmission, making the system versatile across a wide range of robotic applications.

As illustrated in Fig. 2(b), each finger of the manipulator is equipped with a sensor pad containing 256 sensing units (a 16 by 16 sensor array). The size, density, and spatial sensing resolution of the sensor pads can be customized; in our current design, the resolution is set at $3mm^2$ per sensor point. Similar to [11], the tactile sensing pads leverage a triple-layer design, where a piezoresistive layer (Velostat) is sandwiched between two sets of orthogonally aligned conductive yarns serving as electrodes. These layers are then encapsulated between two shaped Polyimide films using a high-strength adhesive (3M 468MP), ensuring robust electrical contact between the electrodes and the piezoresistive film, which is crucial for reliable signal acquisition. The sensor characteristics are repeatable across multiple

devices and reliable over multiple cycles. More detailed information on tactile sensor manufacturing is available in the Appendix.

The resistance of the piezoresistive layer changes in response to applied pressure, enabling each sensor point to convert mechanical pressure into an electrical signal. These analog signals are captured by an Arduino Nano and transmitted to a computer via serial communication. We use a customized electrical readout circuit to acquire data at a frame rate of up to approximately 32.2 FPS. The total cost of one sensor pad and the reading board (without Arduino) is about $20. **We are committed to releasing comprehensive tutorials for hardware manufacturing.**

**Integration of Flexible Tactile Sensor and Soft Gripper.** We install our tactile sensors on the surface of a fully 3D-printed soft gripper made from TPU material (Fig. 2(b)). The tactile sensor pads integrate well with the flexible soft gripper. Our new gripper design offers several advantages. First, the soft nature of the gripper significantly increases the contact area between the sensors and the target objects. This not only helps stabilize the manipulation process but also ensures consistent reflection of the contacting patterns and geometry of the objects. Second, while our visuo-tactile policy provides certain levels of action compliance, the softness of the gripper adds mechanical compliance [42], enabling us to handle fragile objects more effectively.

### 3.2 Multi-Modal Sensing and Teleoperation Setup

We employ a bimanual teleoperation system using two master robots to control two puppet robots [16]. During the data collection process, we gather synchronized multimodal information at a consistent frequency of 10 Hz from various sensors, including tactile sensors, multi-view RGBD cameras (Realsense), and data on robot target actions and current joint states. Synchronization among the sensing information is critical for maintaining the temporal consistency of the multimodal dataset, allowing for accurate alignment between tactile feedback and visual data. We also implement real-time tactile information feedback by visualizing it on the screen, as shown in Fig. 2(a). This enables the human operators to assess the adequacy of contact for secure grasping, which enhances the quality of the collected data, as will be detailed in Sec. 5.

## 4 Learning Visuo-Tactile Dexterity

### 4.1 Problem Formulation

In our study, we address the challenge of learning contact-rich robot skill trajectories through imitation learning. Fig. 3 provides an overview of the integration of visuo-tactile data and the subsequent action generation processes. Specifically, we introduce a visuo-tactile policy, denoted as $\pi : \mathcal{O} \to \mathcal{A}$. This policy maps combined visual and tactile observations $o \in \mathcal{O}$ to actions $a \in \mathcal{A}$. Our method consists of two critical parts: **(1) Dense Visuo-Tactile Representation:** Fig. 3(b) shows the integration of visual and tactile data within a unified coordinate system, which includes: (i) 3D Visual Point Cloud (visualized by •): Captured by the camera, formatted as $P_t^{\text{visual}} \in \mathbb{R}^{N_{\text{vis}} \times 4}$, including an additional empty channel to match the shape of the tactile data. (ii) 3D Tactile Point Cloud (visualized by •): This tactile point cloud includes all the points of the tactile sensing units and uses the sensing value as a feature channel, formatted as $P_t^{\text{tactile}} \in \mathbb{R}^{N_{\text{tac}} \times 4}$. **(2) Policy Learning:** Fig. 3(c) indicates the imitation learning process. Conditioned on our 3D dense visuo-tactile representation, we leverage the diffusion policy [13] to generate actions as a sequence of robot joint states.

### 4.2 Dense Visuo-Tactile Representation

In our approach, instead of separately processing tactile and visual modalities for feature extraction [14], we integrate tactile and visual data by projecting them into the same 3D space. As illustrated in Fig. 3(b), the top row demonstrates the processing of visual observations, while the bottom row depicts the processing of dense 3D tactile points using tactile signals and robot proprioception.

**3D Visual Point Cloud.** We implement a series of data preprocessing procedures on the point cloud captured by the camera, denoted as $P_t^{\text{visual}} \in \mathbb{R}^{N_{\text{vis}} \times 4}$. The process involves four steps: (i) Merge: We combine point clouds from multi-view depth observations to ensure comprehensive coverage of the observed environment. (ii) Crop: The point cloud is cropped to the designated work region using a

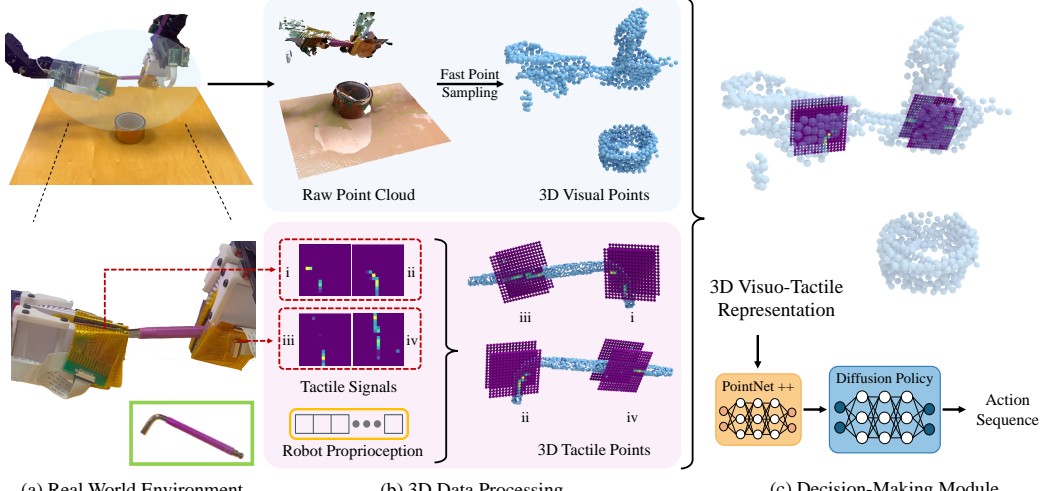

| (a) Real World Environment | (b) 3D Data Processing | (c) Decision-Making Module |

Figure 3: **Visuo-Tactile Policy. Part (a)** shows the real-world setup and the manipulated objects. **Part (b)** illustrates the processing of visual data (upper block) and tactile data (bottom block), followed by their integration within the same 3D coordinates. From the visualization of the tactile signals, depending on the relative movements of the two grippers, the force patterns on the two fingers of the same gripper can differ even when grasping a symmetric part of the tool. Such nuanced information is particularly important for in-hand object manipulation. **Part (c)** outlines our decision-making process, where our network takes the integrated 3D visuo-tactile representations as input and outputs the predicted action sequence.

manually defined bounding box. (iii) Down-Sample: To enhance the efficiency of our visual data processing, we down-sample the point cloud using Farthest Point Sampling (FPS) [63] to ensure a more uniform coverage of the 3D space (compared to uniform sampling). Here, we set $N_{\text{vis}} = 512$ to maintain a balance between geometric detail and computational efficiency. (iv) Transform: The point cloud is transformed into the robot's base frame.

**3D Tactile Point Cloud.** The tactile-based point cloud, $P_t^{\text{tactile}} \in \mathbb{R}^{N_{\text{tac}} \times 4}$, represents the positions and the continuous tactile readings from the tactile units in 3D space. To determine the position of each sensor, we calculate the real-time grippers' positions using forward kinematics based on the robot's joint states. We set $N_{\text{tac}} = 256 \times N_{\text{finger}}$ for the tactile point cloud, where $N_{\text{finger}}$ represents the number of robot fingers equipped with the tactile sensor pads. Each sensor pad consists of 256 tactile points. We set $N_{\text{finger}} = 2$ for the single arm task and $N_{\text{finger}} = 4$ for the bimanual task.

**3D Visuo-Tactile Points.** We then integrate the two types of point clouds, $o = P_t^{\text{tactile}} \cup P_t^{\text{visual}}$, into the same spatial coordinates, as visualized in Fig. 3(c). Each point is also assigned a one-hot encoding to indicate whether it is a visual point or a tactile point. This unified 3D visuo-tactile representation provides the policy network with a detailed and explicit accounting of the spatial relationships between tactile and visual data. This integration introduces an inductive bias that enhances the effectiveness of manipulation in contact-rich tasks, particularly those requiring a comprehensive consideration of both modalities.

### 4.3 Training Procedure

As shown in Fig. 3(c), the decision module in our method is formulated as a conditional denoising diffusion model [64]. It uses the PointNet++ [65] architecture as the backbone and is conditioned on the 3D visuo-tactile representation $o$ to denoise random Gaussian noise into the actions $a$.

## 5 Experiments

### 5.1 Tactile Sensor Hardware Characterization

We evaluated the physical characteristics of our tactile sensors to better understand their performance and signal range. As depicted in Fig. 2(c), the top figure illustrates how the reading signals of our tactile sensors vary with the applied force. We selected 10 sensor units from one sensor pad, and the data from each of them were used to fit individual curves. The bottom figure in Fig. 2(c) shows the

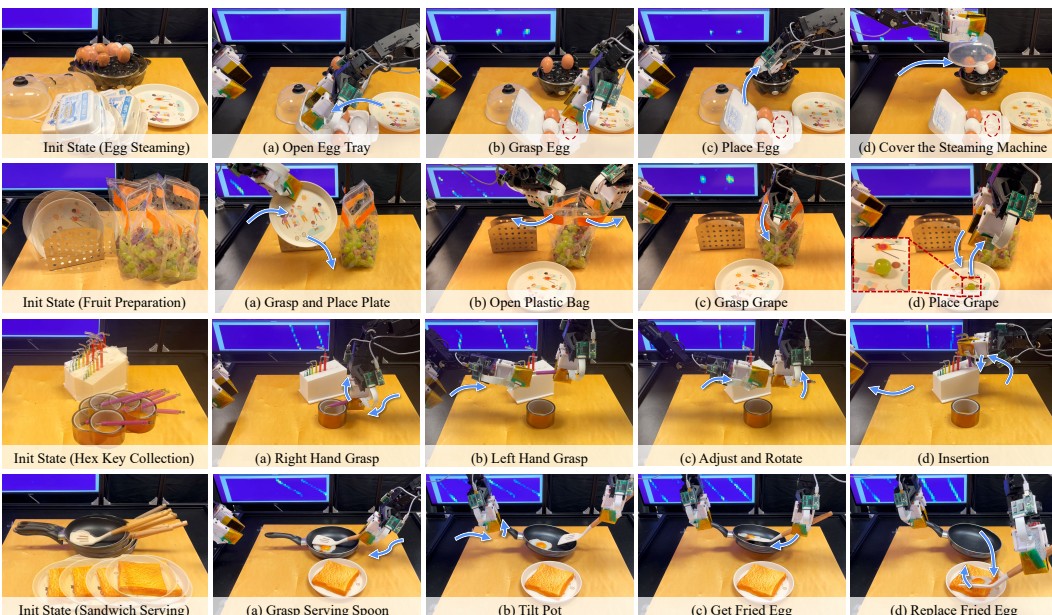

Figure 4: **Policy Rollout.** We evaluate our visuo-tactile policy across four long-horizon, precise manipulation tasks. Detailed descriptions and metrics for these tasks can be found in Sec. 5.2. The first two rows emphasize tasks that require fine-grained force information, while the last two rows focus on tasks that require in-hand object state information. Please check videos on our website for more details.

consistency of our tactile sensor pad under different loads. We divided our sensor into an $8\times8$ grid and summed the readings from each unit. The consistency of the $8\times8$ grid under different loads is shown using a box plot. To demonstrate the capabilities of our dense, continuous tactile sensors, we also conducted additional experiments for 6 DoF pose estimation using only 3D tactile information and models of the manipulated objects. More details can be found in the Appendix.

## 5.2 Experiment Setup for Imitation Learing

We evaluate our multi-modal sensing and learning system on four challenging real-world robotic tasks, each divided into four steps for a more fine-grained assessment of performance (Fig. 14). Tasks are categorized based on how they benefit from the incorporation of tactile signals. Below are the basic descriptions and evaluation metrics for all tasks:

*(1) Tasks Requiring Fine-Grained Force Information:*

**Egg Steaming**. The robot uses its right hand to open the egg tray first. Then the robot must grasp and place an egg into an egg cooker. Subsequently, the left hand is used to relocate and secure the cooker's cover over the egg. *Evaluation Metrics:* The task is considered successful if the egg is placed without damage and the cooker's cover is properly positioned over the egg. A handling failure occurs if the egg falls due to insufficient grip force or cracks under excessive force.

**Fruit Preparation**. The robot uses its left hand to grasp the plate and place it on the table. Subsequently, both robot arms collaborate to open the plastic bag. Then, the right arm grasps a grape or several grapes and places them on the plate. *Evaluation Metrics:* The task is considered successful if the grapes are placed on the plate without sustaining any damage.

*(2) Tasks Requiring In-Hand State Information:*

**Hex Key Collection:** The right hand is required to grasp the Hex Key, and then the left hand needs to grasp the Hex key followed by an in-hand adjustment of the Hex Key's position using its left hand. Subsequently, the robot is required to accurately insert the Allen wrench into the hole in the box. *Evaluation Metrics:* A successful insertion of the Allen wrench into the hole is considered a successful operation. Additionally, any failure to properly adjust the Hex Key's position may result in the inability when inserting it into the hole.

Tasks Requiring Fine-Grained Force Information

| Modalities | Egg Steaming (30 demos) | | | | Fruit Preparation (30 demos) | | | |
|---|---|---|---|---|---|---|---|---|
| | Open Egg Tray | Grasp Egg | Place Egg | Whole Task | Grasp and Place Plate | Open Plastic Bag | Grasp Grapes | Whole Task |
| RGB Only | 0.95 | 0.75 | 0.60 | 0.50 | 0.95 | 0.75 | 0.45 | 0.45 |
| RGB w/ Tactile Image | 0.95 | **0.90** | 0.80 | 0.70 | 0.90 | 0.80 | 0.70 | 0.70 |
| PC. Only | **1.00** | 0.75 | 0.65 | 0.55 | **1.00** | 0.85 | 0.50 | 0.45 |
| PC. w/ Tactile Image | **1.00** | 0.85 | 0.70 | 0.70 | 0.95 | **0.90** | 0.75 | 0.65 |
| PC. w/ Tactile Points (Ours) | **1.00** | **0.90** | **0.90** | **0.85** | 0.95 | 0.85 | **0.80** | **0.80** |

Tasks Requiring In-Hand State Information

| Modalities | Hex Key Collection (30 demos) | | | | Sandwich Serving (50 demos) | | | |
|---|---|---|---|---|---|---|---|---|
| | Right Hand Grasp | Left Hand Grasp | In-Hand Adjustment | Whole Task | Grasp Serving Spoon | Tilt Pot | Get Fried Egg | Whole Task |
| RGB Only | 0.90 | 0.85 | 0.55 | 0.45 | **1.00** | 0.95 | 0.70 | 0.60 |
| RGB w/ Tactile Image | 0.95 | 0.90 | 0.70 | 0.60 | **1.00** | **1.00** | 0.85 | 0.75 |
| PC. Only | **1.00** | 0.90 | 0.65 | 0.65 | **1.00** | **1.00** | 0.75 | 0.65 |
| PC. w/ Tactile Image | 0.95 | 0.85 | 0.60 | 0.50 | **1.00** | 0.90 | 0.75 | 0.65 |
| PC. w/ Tactile Points (Ours) | **1.00** | **0.95** | **0.95** | **0.90** | **1.00** | **1.00** | **0.90** | **0.85** |

Table 1: **Comparison with Baselines.** We evaluate our policy over 20 episodes and the best performance for each task is bolded. Please check our website for more comparison videos.

**Sandwich Serving:** First, the right hand is required to grasp the serving spoon. Second, the left hand needs to hold the pot handle and then tilt the pot. The right hand should then retrieve the fried egg from the pot and serve it on the bread. *Evaluation Metrics:* The robot must successfully retrieve the fried egg from the pot and serve it on the bread.

In the experiments, we primarily compare our methods with the following baselines. We trained all policies for 2,000 epochs. All methods, including ours and the baselines, use three camera views.
*(1) RGB Only.* This method uses multi-view RGB from cameras as input for the image-based diffusion policy. We use the same implementation as [13].
*(2) RGB w/ Tactile Image.* This method processes multi-view RGB images and tactile images through different branches for the diffusion policy. We use a CNN as the feature extractor for tactile images.
*(3) PC. Only.* This method uses only multi-view visual point clouds as the sensing modality for the diffusion policy. We use PointNet++ as the feature extractor.
*(4) PC. w/ Tactile Image.* This method fuses multi-view visual point clouds and tactile images through different branches for the diffusion policy. We use a CNN as the feature extractor for tactile images and PointNet++ as the feature extractor for point clouds.

### 5.3 Qualitative Analysis

Our policy integrates three modalities: vision, tactile sensing, and robot proprioception, and has proven to be effective through four challenging, long-horizon tasks. We observe three key benefits of integrating touch. (1) *Tactile sensors provide critical feedback on the presence of contact and the appropriate amount of force to apply.* For example, a common issue observed with the baseline during the egg-steaming task is the application of insufficient force, which often results in the egg falling or not being successfully grasped from the tray. Similarly, during the grape-handling task, a typical error occurs when the gripper attempts to grasp multiple grapes at once, applying excessive force that damages the fruit. (2) *Our policy leverages detailed contact patterns provided by touch to address visual occlusions effectively.* A common failure in the baseline during the Hex Key collection task is the inability to adjust the hex key's position in hand, leading to failures in subsequent insertion attempts. In the sandwich serving task, tactile feedback is crucial for understanding the in-hand states and the spatial relationship between the contact area and all tools, especially when tools may passively rotate as the spoon interacts with the pot—a frequent cause of failures in baseline approaches. (3) *Tactile feedback provides the confidence needed to transition between task stages.* This is especially important in scenarios where the visual information is noisy or highly occluded. During our experiments on grasping grapes in a bag, the visual-only policy frequently failed, often getting stuck in the bag and unable to progress to the next stage. **More details and videos about failure cases can be found in the Appendix.**

**Ablation Study on Varying Levels of Visual Information.** Visual occlusion presents a significant challenge in manipulation tasks, especially in bimanual tasks. To determine whether tactile sensing can compensate for visual occlusion, we varied the levels of occlusion by changing the number of cameras used both in the training set and during the policy rollout. As detailed in Table 2, *Cam* refers

| Settings | Egg Cooking (30 demos) | | | | Hex Key Collection (30 demos) | | | |
|---|---|---|---|---|---|---|---|---|
| | Open Egg Tray | Grasp Egg | Place Egg | Whole Task | Right Hand Grasp | Left Hand Grasp | In-Hand Adjustment | Whole Task |
| Tactile Points Only | 0.70 | 0.55 | 0.10 | 0.10 | 0.50 | 0.30 | 0.30 | 0.15 |
| Single Cam PC. Only | 0.90 | 0.65 | 0.60 | 0.50 | 0.80 | 0.60 | 0.45 | 0.30 |
| Multi Cam PC. Only | **1.00** | 0.75 | 0.65 | 0.55 | **1.00** | 0.90 | 0.65 | 0.65 |
| Single Cam PC. w/ Tactile Points | 0.95 | **0.90** | 0.85 | 0.80 | 0.85 | 0.80 | 0.75 | 0.70 |
| Multi Cam PC. w/ Tactile Points | **1.00** | **0.90** | 0.90 | **0.85** | **1.00** | **0.95** | **0.95** | **0.90** |

Table 2: **How Touch Complement Varying Levels of Visual Information?** We manipulate visual occlusion by varying the number of cameras, thereby testing how tactile feedback compensates for reduced visual input.

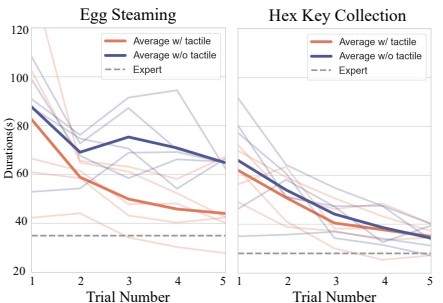

Figure 5: **Tactile Feedback Improves the Demonstration Data Quality.** Users new to the system perform better with both visual and tactile feedback than with vision alone.

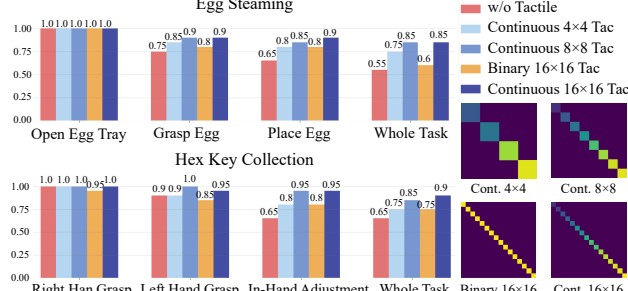

Figure 6: **Comparison of Varying Types of Tactile Information.** Across all tested tactile modalities, we found that denser and continuous signals consistently outperform all other comparison groups. *Tac* refers to the use of tactile signals.

to the use of visual point clouds, while *Tactile Points* denotes the use of tactile point clouds in 3D space, aligned with the camera coordinates. Results show that even as visual occlusion increases with fewer cameras, the policy continues to perform well with the aid of tactile information.

**Ablation Study on Varying Levels of Tactile Information.** Contact data can vary in form and resolution. Dense and continuous tactile data not only indicate the presence of contact but also measure the amount of applied force and the detailed local contact pattern—capabilities that sparse binary tactile signals lack [26]. To validate this, we varied the tactile resolution from 16×16 to 8×8 and 4×4 and added an additional baseline that only considers binary signals. All baselines and ours in Fig. 6 incorporate the multi-view point clouds. We found that continuous signals significantly enhance performance, especially in actions requiring precise in-hand orientation of the grasped objects. Dense tactile patterns at a resolution of 16×16 slightly outperform the 8×8 resolution and noticeably surpass the 4×4 and binary comparison groups.

**Demonstration Data Quality with Tactile Feedback.** As shown in Fig. 2(a), real-time tactile signals are visualized and displayed on the operator's screen during data collection. This feature not only enhances the efficiency of data collection but also improves the quality of the data. To validate the effectiveness of tactile feedback, we enlisted 10 new users to complete two tasks. Five of these users operated the teleoperation system with tactile feedback, while the other five did not. The duration to complete the tasks is shown in Fig. 5.

## 6 Conclusion and Limitation

**Conclusion.** In this paper, we develop a multi-modal sensing and learning system for contact-rich robotic manipulation. We introduce a dense, flexible tactile sensor array that covers a larger area of thin, soft robot grippers. We also propose a unified 3D visuo-tactile representation, which explicitly accounts for the 3D structure and spatial relationship between vision and touch. We demonstrate the effectiveness of these innovations in a range of challenging manipulation tasks. **We are committing to release our code and hardware setup.**

**Limitations.** We acknowledge that collecting real data with a multi-modal perception system is expensive, due to the data collection itself and the addition of tactile sensing modality. One missing piece of our study is the simulation of our tactile sensor, limiting our ability to introduce randomization or scale up the data. In future work, we aim to enhance the system's capabilities to better exploit advances in physical simulation, further improving the generalizability and robustness of the policy.

# 7 Acknowledgement

The Toyota Research Institute (TRI) partially supported this work. This article solely reflects the opinions and conclusions of its authors and not TRI or any other Toyota entity.

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

# Supplementary Materials

## Contents

# A  Tactile Sensor Hardware

## A.1  Tactile Sensor Manufactory

### A.1.1  Tactile Sensor Pad Design

The tactile sensing pads leverage a triple-layer design, where a piezoresistive layer (Velostat) is sandwiched between two sets of orthogonally aligned conductive yarns serving as electrodes. During the tactile sensor manufacturing, we first align 16 Stainless Thin Conductive Threads on top of the Velostat layer and then use high-strength adhesive (3M 468MP) to ensure robust electrical contact between the electrodes and the Velostat layer. Additionally, we use adhesive to secure the conductive thread connections to the connector. The connector links all the threads to a flexible flat cable, allowing the signal to be transmitted to the PCB board. This design makes the wires of our tactile sensor highly flexible, facilitating easier installation in various locations, such as the robot end-effector, which requires constant movement during manipulation. To ensure the tactile sensor's long-term robustness, we attach a polyimide layer on top of the adhesive. Polyimides are known for their thermal stability, good chemical resistance, excellent mechanical properties, and characteristic

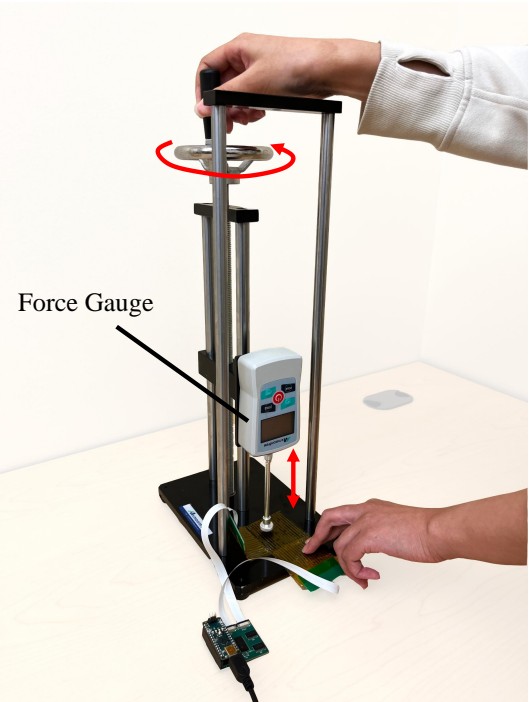

Force Gauge

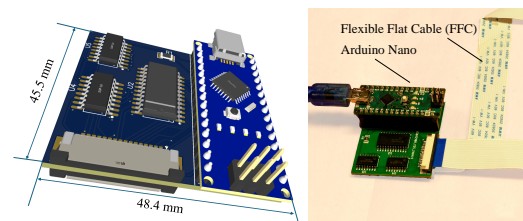

Flexible Flat Cable (FFC)
Arduino Nano

Figure 8: **Tactile Reading Board Design.**

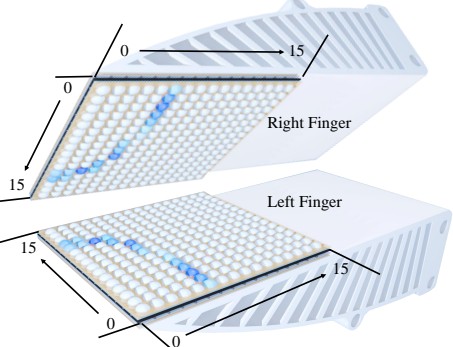

Figure 7: **Tactile Physical Characteristics Evaluation Experiment**

Figure 9: **Tactile Sensor Order Visualization.**

orange/yellow color. After completing these steps, we finish aligning the 16 threads for the rows. Then, we flip the sensor and align the 16 threads for the columns.

After obtaining the tactile sensor pad, we attach the sensors to the robot fingers. The order of each tactile sensor unit is visualized in Fig. 9. We clearly define the tactile order to ensure that each sensor's position can be accurately calculated, and the tactile signals can correctly correspond to our real setting and dataset.

### A.1.2 Reading Board Design

To ensure easy installation of the tactile reading board in the robot, we have designed it to be as compact as possible, as shown in Fig. 8. The tactile reading board measures 45.5 mm × 48.4 mm and includes an Arduino. The small size further enhances the scalability of our tactile sensors. We use two 8-bit shift registers and one 16-channel analog switch to process the tactile signals, which are then input to the Arduino. The ADC in the Arduino converts the analog signals from the tactile sensor into digital signals and forwards them to the host via serial communication. We will release a comprehensive reading board scheme so that the community can directly order from a PCB supplier to easily replicate our tactile sensor.

### A.2 Tactile Hardware Evaluation Experiment

### A.2.1 Physical Characteristics

To investigate the physical characteristics of our tactile sensors, we designed two experiments. As illustrated in Fig. 7, we use a force gauge to apply specific force on the tactile sensor surface. The first experiment tests how individual tactile sensor units react to applied force. The second experiment aims to test the consistency of the entire sensor pad, showcasing the variance between different regions on the tactile sensor pad.

**Individual Sensor Performance.** We began by randomly selecting 10 sensors from a total of 256 sensors in one sensor pad. For each selected sensor, we applied a normal force incrementally, ranging

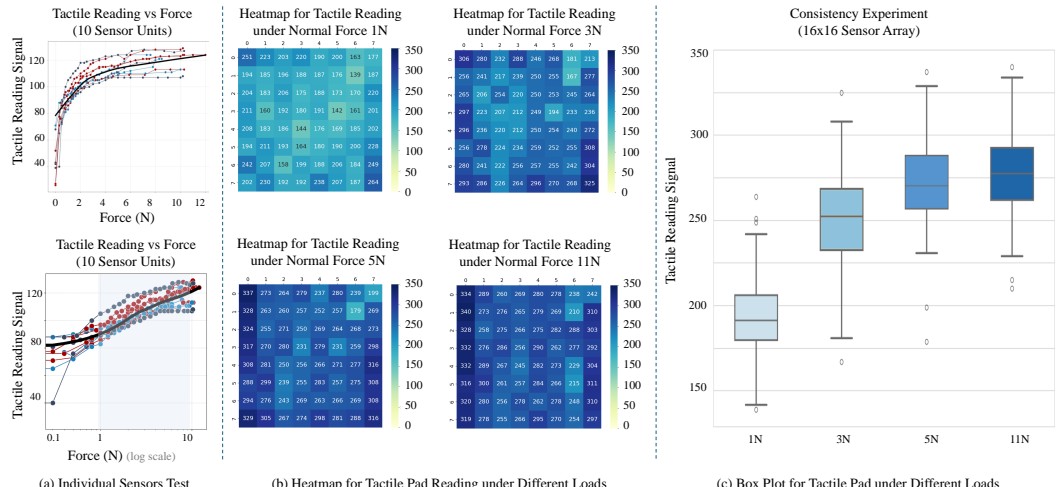

Figure 10: **Results of Physical Characteristics Experiments. Part (a)** shows the results of individual sensors' performance according to the force applied to their surface. **Part (b)** demonstrates the tactile sensor pad's consistency under different normal forces. Each heatmap displays the tactile sensor pad's readings in an $8 \times 8$ grid, where each number represents the sum of four sensor units. **Part (c)** presents the results from part (b) in a single figure, illustrating the mean and standard deviation.

from 0 to 12 N, and recorded the stable tactile reading accordingly. Each sensor generated an average of 24 data points. This method allowed us to observe the individual sensor's response to varying force levels and identify its saturation thresholds. As shown in Fig. 10 (a), we plot tactile reading versus normal force and identified that the saturation zone begins when the normal force exceeds 9 N. The fitting curve for the 10 sensors is depicted in the black line. Additionally, we applied a logarithmic scale to the x-axis (normal force), resulting in an approximately linear region for normal forces from 1 N to 9 N. The region is highlighted with a blue background, as illustrated in Fig. 10 (a).

**Tactile Sensor Pad Consistency.** In the second part of the experiment, we used a $16 \times 16$ tactile sensor pad and divided it into $8 \times 8$ blocks, with each block comprising 4 sensors ($2 \times 2$ matrix area). Uniform loading was applied across each block using the force gauge with a circular contacting area of 176.7 mm$^2$. For each $2 \times 2$ block, we collected the sum of the four tactile readings from individual sensors, enabling us to generate a heat map that visualizes the sensor response under specific loading conditions across the entire pad. Four different loading conditions (1 N, 3 N, 5 N, and 11 N) were applied to comprehensively assess the overall performance, providing a detailed representation of the resolution under varying forces. For each force condition, we measured once for each block, resulting in a total of 64 data points per condition. We then generate a heatmap for each force condition as shown in Fig. 10 (b). We calculated the mean and standard deviation for these data points and removed outliers. Finally, as illustrated in Fig. 10 (c), we generated a box plot from 4 sets of 64 tactile readings, demonstrating its consistency across the entire sensor and the stable functionality of the tactile sensors.

### A.2.2    6-DoF Object Pose Estimation

In the main paper, we demonstrated the effectiveness of dense, continuous tactile information for fine-grained manipulation tasks. To gain a more comprehensive understanding of the information captured by our proposed sensors, we conducted additional experiments on 6-DoF object pose estimation. These experiments revealed that the sensors embed information about object geometry and local contact patterns, which is crucial for manipulation tasks requiring robust and adaptive grasping as well as precise in-hand reorientation behavior.

Specifically, we define the task as estimating the 6-DoF pose of an object using only tactile observations, **without** any visual input. We assume that the object geometry is known and denote its 3D point cloud as $P^{\text{obj}} \in \mathbb{R}^{N \times 3}$. The tactile observation, obtained by filtering the tactile-based point cloud according to the activation value, is denoted as $P^{\text{tactile}} \in \mathbb{R}^{M \times 3}$. Our objective is to track the

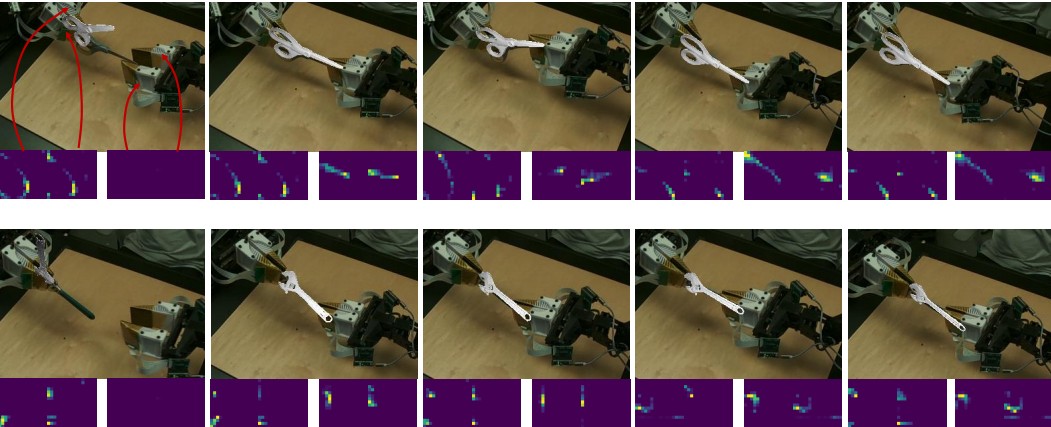

Figure 11: **Pose Estimation.** In this experiment, we estimate the object pose **without** vision information. We can see that our pose estimation becomes more accurate as we have more complete tactile signals. We can also track the object's pose as it rotates. Through this estimation, we demonstrate that our hardware can be potentially used for in-hand pose estimation and other visuotactile tasks.

pose of the object in the 3D space, $\mathbf{T} \in \mathbb{SE}(3)$, where,

$$\mathbf{T} = \begin{bmatrix} \mathbf{R} & \mathbf{t} \\ 0^T & 1 \end{bmatrix} \in \mathbb{SE}(3), \tag{1}$$

in which the Euclidean group $\mathbb{SE}(3) := \{\mathbf{R}, \mathbf{t} \mid \mathbf{R} \in \mathbb{SO}^3, \mathbf{t} \in \mathbb{R}^3\}$.

We solve the pose-tracking problem using particle filtering [66]. We first define our observation function $P^{\text{obs}} = f(\mathbf{T})$ and then the weighting functions $w = g(P^{\text{obs}}, P^{\text{tactile}})$ as follows:

$$f(\mathbf{T}) = \mathbf{R}P^{\text{obj}} + \mathbf{t},$$
$$g(P^{\text{obs}}, P^{\text{tactile}}) = \sum_{p_i \in P^{\text{tactile}}} \min_{p_j \in P^{\text{obs}}} ||p_i - p_j||^2. \tag{2}$$

The observation function transforms the object model point cloud using $\mathbf{T}$, while the weighting function calculates the distance from the contact points to the observation points. In practice, we scale the weights using an exponential function to facilitate convergence. Given the observation and weighting functions, we employ a standard particle filter to determine the object's pose.

Some example results are shown in Figure 11. Before the right-side robot makes contact with the object, we can only rely on the tactile signals from the left-side robot. Therefore, there are a lot of plausible solutions. Although our estimated pose is one of the plausible solutions given the one-side tactile signal, the estimation is still inaccurate. When the right-side robot contacts the object, our estimated pose aligns well with visual observation. Also, when the object is rotating in the hand, the object pose is tracked accurately.

## B  Experiment details for Imitation Learning

### B.1  System Overview

As shown in Fig. 12, We employ a bimanual teleoperation system with three Realsense cameras and four tactile sensor pads on four robot fingers. Our tactile signal communication is facilitated by a multi-threaded ROS (Robot Operating System) node. This node captures tactile signals and publishes them at a frequency of 30 Hz. All data, including that from cameras and tactile sensors, is collected through multi-threading. Each data frame received is timestamped, and after an episode is completed, we align all data with these timestamps. This synchronization is crucial for maintaining the consistency of the multimodal dataset, enabling accurate temporal alignment between tactile feedback and visual data. To manage the heavy load of processing frames from three cameras, we

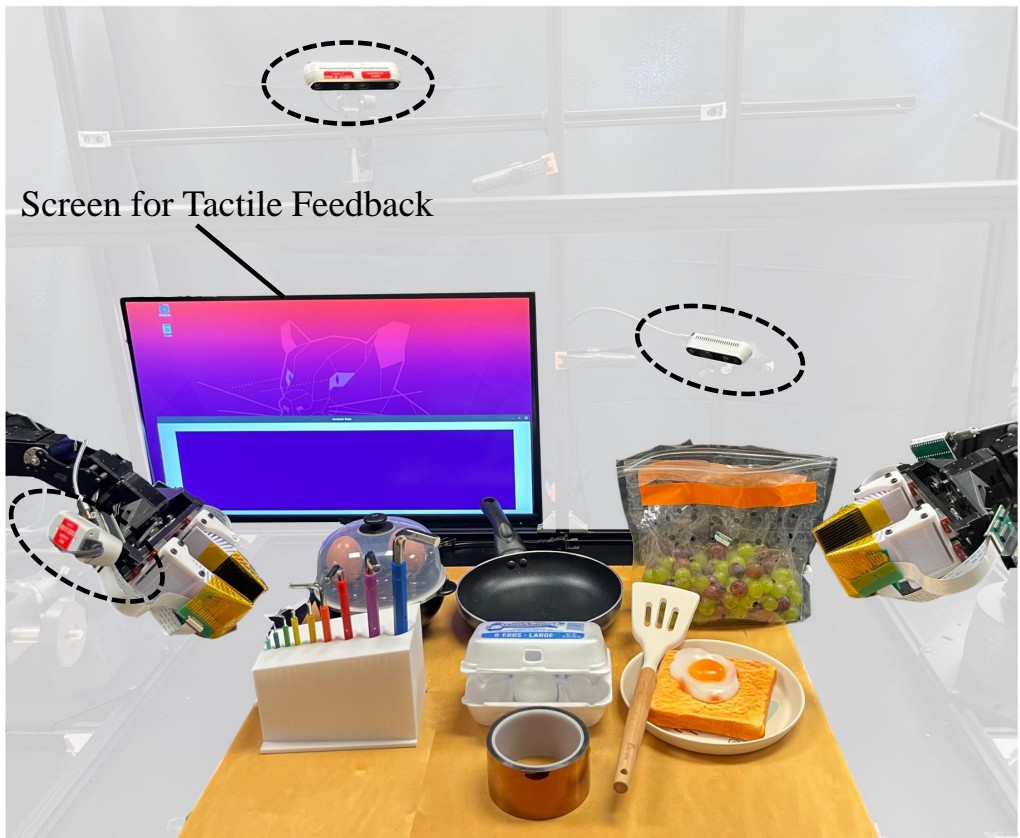

Figure 12: **System Overview.** We attach four tactile sensors to four robot fingers and install three Realsense cameras to cover the workspace. All the objects used for the task are shown in the workspace. Additionally, we install a background screen to display the tactile feedback.

collect data at 10 Hz to ensure consistency. We set a top camera (Realsense 455) to cover the entire workspace and positioned two other cameras (Realsense 435) close to the workspace to capture more detailed information. When using point cloud data from multiple cameras, we incorporate data from all cameras. For the baseline method using a single camera, we use only the top camera.

We also implement real-time tactile information feedback, as shown in Fig. 12 (a). During data collection, tactile signals are visually displayed on the operator's screen, enabling them to assess the adequacy of contact for secure grasping. Additionally, during the policy rollout, this visualization helps us see in real-time how tactile information relates to robot motion.

## B.2 Experiment Setup Details

In this section, we discuss the detailed information of the four tasks described in the main paper. Each task consists of four steps, as illustrated in Fig 14. We will discuss the motions and evaluation metrics for each step, and highlight how these steps demonstrate the capabilities of our tactile sensors. The typical failure cases are shown in Fig. 13 and will be discussed in the following sections.

### B.2.1 Details for the Egg Steaming task

*Step 1: Open Egg Tray.* The robot uses its right hand to open the egg tray, which mirrors the common scenario where the egg is often occluded by the tray. This realistic setup is maintained to reflect daily life, avoiding task simplification. *Evaluation Metrics:* The robot must open the tray sufficiently to allow its fingers to grasp the egg. Failure to open the tray adequately will result in the subsequent task failing. The initial position of the egg tray will be randomized within an area of 7-10 cm during both data collection and policy rollout.

*Step 2: Grasp Egg.* The robot uses its right hand to grasp the egg in the tray. This motion is complex, requiring the robot to slowly increase the force and carefully grasp the egg despite heavy occlusion. The robot with a visuo-tactile policy will retry if there is no stable tactile signal in hand, while a vision-only policy may proceed to the next goal due to heavy occlusion as shown in Fig. 13 (a). *Evaluation Metrics:* The robot can reattempt to grasp the egg, but the step fails if it moves to the next stage without the egg or if the egg falls during the transition from the tray to the steaming machine. Additionally, prolonged time spent in the egg tray will also be considered a failure.

*Step 3: Place Egg.* The robot needs to safely place the egg in the steaming machine, which already contains two eggs. It must avoid causing the other eggs to fall while placing the egg in-hand. This step highlights our flexible thin sensor's capability to perform fine-grained tasks in narrow spaces. As the robot hand exits the steaming machine, tactile information ensures there is no contact between the egg and the gripper, signaling the robot to proceed to the next stage. In contrast, a vision-only policy may cause confusion about whether the robot can move out safely, potentially prolonging its stay in the steamer and increasing the risk of dislodging the other eggs. *Evaluation Metrics:* The robot can place the egg anywhere inside the steaming machine, but the step fails if the robot does not place the egg in the steaming machine or if it causes the other eggs to fall to the ground.

*Step 4: Cover the Steaming Machine.* The robot needs to use its left hand to grasp the cover of the steaming machine and place it safely inside. This task is challenging due to the unique shape of the steaming machine's handle, as shown in Fig. 13 (a). The robot must apply a precise amount of force to the handle: sufficient to lift it but not so much that the cover flips and falls to the ground. The robot must apply a precise amount of force to the handle: sufficient to lift it but not so much that the cover flips and falls to the ground. This step showcases how our tactile sensor enables the robot to perform fine-grained grasping manipulations, similar to a human's ability to apply suitable and stable force to grasp objects. *Evaluation Metrics:* The robot is allowed multiple attempts to grasp the cover. The task is considered successful if the cover is securely placed on the steaming machine. It is considered a failure if the cover flips or falls during the process.

### B.2.2 Details for the Fruit Preparation Task

*Step 1: Grasp and Place the Plate.* The robot needs to use its left hand to grasp and place the plate on the table. This step introduces additional randomization and variance due to the varying positions of the plates, increasing the task's complexity. *Evaluation Metrics:* The task is considered successful if the robot grasps the plate and places it on the table.

*Step 2: Open Plastic Bag.* The robot needs to use its two hands to cooperate together to open the bag. The plastic bag is transparent and usually adds additional noise to the point cloud. *Evaluation Metrics:* The task is considered successful if the robot opens the bag wide enough for the gripper to get in.

*Step 3: Grasp the fruit.* The robot needs to use its right hand to get inside the plastic bag and grasp the fruit. This step is the most important and difficult in this task. First, as shown in Fig. 14(Task 2: Fruit Preparation), the robot and manipulated objects are highly occluded in the bag, making it impossible for visual information to observe critical details. Our visuo-tactile policy will grasp multiple times until there is stable tactile information to secure the grapes, while a vision-only policy typically attempts the motion once regardless of the presence of grapes, making the grasping success a random event. Second, the grapes are usually clustered together, requiring the robot to apply suitable force to avoid damaging the fruit. Our visuo-tactile policy can successfully grasp single or multiple grapes from the bag, while a vision-only policy may break the grapes when the robot grasps multiple grapes (as shown in Fig. 13 (b)). since it aligns the gripper joint states instead of using force-related information. Third, this task also showcases our sensors' human-like dexterous manipulation; our tactile-integrated gripper is thin enough to get into the gaps between grapes, making it easier to grasp the grapes in a cluster. *Evaluation Metrics:* The task is considered successful when the robot successfully grasps the fruit out of the plastic bag. The task is considered a failure if the robot breaks the grapes or moves to the next stage without the grapes. The policy also fails if the robot stays in

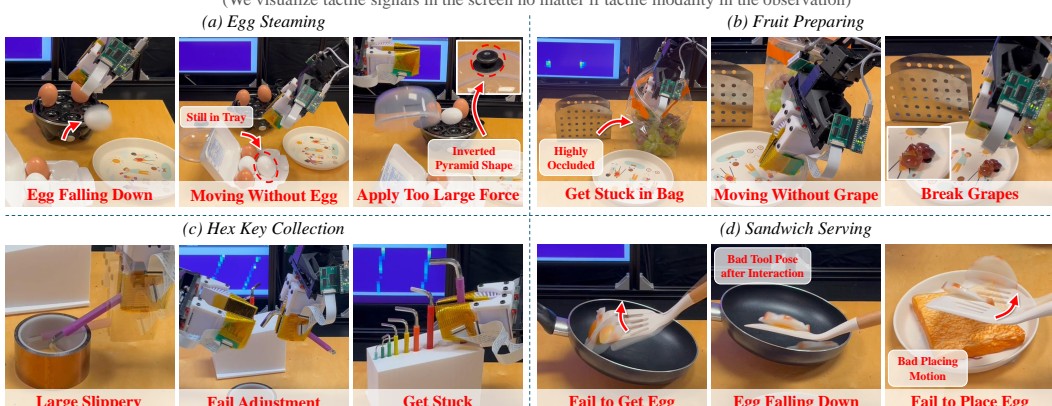

Figure 13: **Failure Cases.** We present typical failure cases of the baseline method for all four tasks and analyze the reasons for these failures to highlight the complexity of the tasks and the importance of tactile feedback during these steps.

the bag for a long time without moving, which usually happens with the vision-only policy that is confused about the states of the objects and the robot end-effector under high occlusion.

*Step 4: Place grape.* The robot needs to place the grapes on the plate. The task may fail if the robot uses too much force to grasp the grapes, causing them to stick in the gripper and resulting in failure. *Evaluation Metrics:* The task is considered successful if the robot successfully places the grapes on the plate and returns to the initial position.

### B.2.3 Details for the Hex Key Collection Task

*Step 1: Right Hand Grasp.* The robot needs to use its right hand to grasp the tail of the hex key and lift it stably to the middle of the air. The initial position of the hex key is tricky, but it reflects a common daily life scenario where only the tail of the hex key is accessible, requiring additional adjustments to insert the hex key properly. A typical failure case of baselines, shown in Fig. 13 (c), occurs when the robot does not secure a stable grasp, resulting in significant slippage during the lifting process. Even if the hex key remains in-hand, this slippage can cause subsequent task failures. One observation during the experiment is the consistent small slippage during the first grasp, leading to variations in the hex key's in-hand pose, which adds complexity to the following steps. *Evaluation Metrics:* The robot successfully grasps the hex key without significant slippage.

*Step 2: Left Hand Grasp.* The robot needs to use its left hand to grasp the head of the hex key to ensure the following adjustment step. *Evaluation Metrics:* The robot left hand successfully grasp the hex key.

*Step 3: In-hand Adjustment.* The robot's left and right hands need to cooperate to adjust the hex key's position so that it is in a ready pose for the following insertion. Our goal is to adjust the hex key to be perpendicular to the robot's fingers, making the subsequent insertion task easier. The vision-only policy usually fails to adjust the position correctly, shown in Fig 13(c), making the following insertion impossible. *Evaluation Metrics:* The robot's two arms must cooperate to adjust the hex key's pose. The final pose should have a sufficiently long tail, and the hex key should be almost perpendicular to the robot's fingers.

*Step 4: Insertion.* This step is complex because the pose of the hex key in hand varies, even if the robot successfully adjusts the hex key's position in the last step. A successful policy can implicitly reference the hex key's position in hand and make the necessary adjustments for insertion. *Evaluation Metrics:* The robot successfully inserts the hex key into the hole rather than placing it on the table or getting stuck during the insertion process.

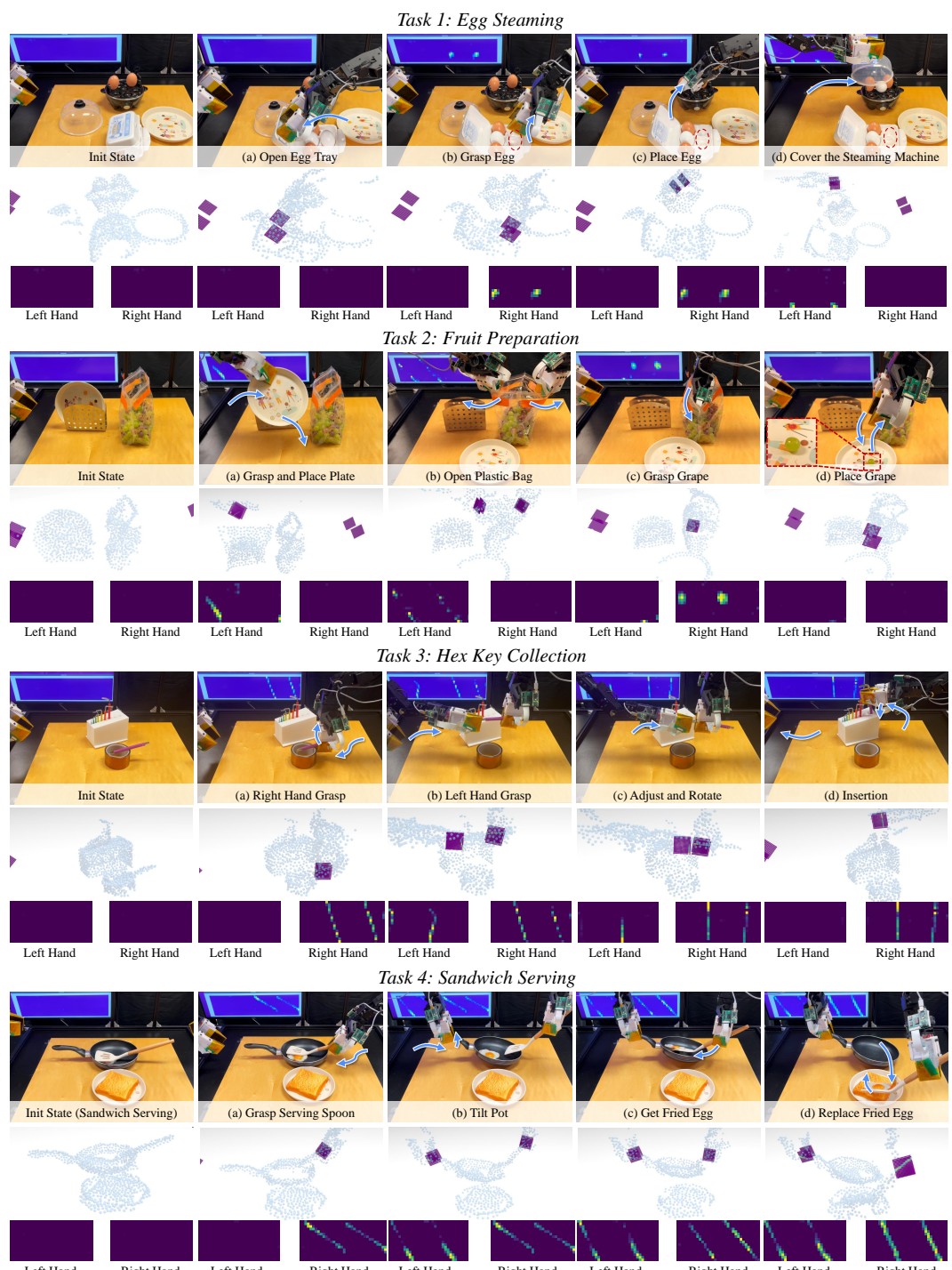

Figure 14: **Quantative Result of Tactile Representation.** Here we showcase a total of four tasks. For each task, the first row presents the real image. In the second row, we visualize our visuo-tactile points in a unified 3D space to demonstrate how tactile points can infer spatial relationships between objects and contact areas. The third row provides a 2D image to clearly visualize the tactile signals.

### B.2.4 Details for the Sandwich Serving Task

*Step 1: Grasp Serving Spoon.* The robot needs to use its right hand to grasp the spoon and lift it into the air. *Evaluation Metrics:* The spoon is successfully lifted into the air with minimal slippage.

*Step 2: Tilt Pot.* In order to successfully obtain the egg in the next step, the robot's left hand needs to grasp the pot's handle and tilt the pot. The gripper should not exert excessive force to ensure that the handle does not rotate within the robot's hand. *Evaluation Metrics:* The robot's left hand must successfully grasp the handle and then tilt it to a certain angle.

*Step 3: Get Fried Egg.* The robot's two hands need to cooperate to retrieve the fried egg. The right hand will use the spoon to reach the bottom of the pot and maneuver beneath the egg. During this process, the spoon will passively rotate in the hand. Our visuo-tactile policy can explicitly track the states of the spoon, while the baseline policy often fails due to the spoon's rotation in the hand. *Evaluation Metrics:* The robot successfully retrieves the fried egg with the spoon.

*Step 4: Replace Fried Egg.* The robot needs to move the spoon to the top of the bread and tilt it to place the egg on the bread. A typical failure occurs when the robot does not perform a successful tilt motion due to changes in the spoon's position within the hand. Our visuo-tactile policy can account for these changes and adjust the motion accordingly. *Evaluation Metrics:* The robot successfully places the fried egg on top of the bread.

### B.3 Single Arm Experiments

**Light Bulb Retrieval:** (i) Rotate. The robot locates the light bulb, rotates it, and retrieves it from the base. (ii) Relocation. The robot places the retrieved light bulb carefully on the table.

**Peg Insertion:** (i) Grasp. The robot grasps a peg from the table. (ii) Insertion. The robot inserts the peg into a hole.

Single-Arm Tasks

| Modalities | Light Bulb Retrieval (30 demos) | | Peg Insertion (30 demos) | |
|---|---|---|---|---|
| | Rotation | Relocation | Grasp | Insertion |
| RGB Only | 0.85 | 0.85 | 1.00 | 0.30 |
| RGB w/ Tactile Image | 0.95 | 0.95 | 1.00 | 0.65 |
| PC. Only | 0.85 | 0.85 | 1.00 | 0.65 |
| PC. w/ Tactile Points (Ours) | **1.00** | **1.00** | **1.00** | **0.85** |
| Ours (Using a New Set of Sensors) | **1.00** | **1.00** | / | / |
| Ours (Changing Sensor Location Slightly) | 0.95 | 0.90 | / | / |

Table 3: **Comparison with Baselines.** We evaluate our single-arm policy over 20 episodes and the best performance for each task is bolded. Our sensors are repeatable across devices after normalization. Even though the reading may have slight variances among different sensors, the force pattern and distribution are almost identical after normalization.

### B.4 Learning Details

We use Pointnet++ as the learning backbone. we employ hierarchical feature extraction and processing for point cloud data. We use three set abstraction layers: the first set abstraction layer processes 64 points with a 0.04 radius and 16 samples using a multi-layer perceptron (MLP) with layers [64, 64, 128]; the second set abstraction layer processes 16 points with a 0.08 radius and 32 samples using an MLP with layers [128, 128, 256]; the third set abstraction layer serves as a global abstraction layer with an MLP of [256, 512, 1024]. For further feature processing, we use fully connected layers: the first fully connected layer transforms 1024 features to 512, and the second fully connected layer reduces 512 features to 256. We disable batch normalization layers.

