# OpenReview forum: "3D-ViTac: Learning Fine-Grained Manipulation with Visuo-Tactile Sensing"
_robot-learning.org/CoRL/2024/Conference — CoRL 2024_

### Official Review · Reviewer_AvTY · 2024-07-21
**CORL 2024 Submission 170 Review**

**Originality:** 4
**Technical Quality:** 5
**Clarity Of Presentation:** 5
**Potential Impact:** 3
**Recommendation:** 3
**Confidence:** 4

**Review:**

The major contribution of the paper seems to be the integration of the tactile sensors in STAG glove [11] to the gripper setting and overall implementation of building and learning from a multi-modal unified point cloud. There is not much originality in the learning aspect other than using continuous tactile values for the point cloud rather than binary values as Yuan et al. [27] did. They are using well-explored methods for the policy architecture. However, this still builds very clearly on top of previous work, and the execution of the overall framework is well done.

The authors conducted thorough experiments for the tactile sensor design and robot experiments for all of their tasks related to the imitation learning part of the paper. Overall, I find the work valuable to share with the community.

The paper is relatively easy to understand, well-written, and the figures are clear and helpful.

**Strengths:**

- Although the tactile sensing mechanism used already exists in the literature, the authors designed new sensors with this mechanism to be used on the grippers. These sensors claim to have characteristics that are repeatable across multiple devices, reliable over multiple cycles, cheap, scalable, and provide dense readings. These features are very valuable in tactile sensing and are, to my knowledge, lacking in the robotics community so far.
- The tasks in the paper are fairly complicated, and the behavior of the model seems to reason very well with the fusion of tactile and vision.
- The authors included a good number of robot experiments to show the importance of different parts of their method. I especially found the experiments that compare different representations of modalities and the success rates with respect to the number of cameras quite insightful.

**Weaknesses:**

- I believe there should be more explanation and motivation on why, as a community, we need better tactile sensors and more experiments on how well the new tactile sensors they introduced are working. Currently, one of the biggest problems with magnetic tactile sensors is that their readings are not transferable between skins and the readings drift with time. The authors claim the readings from the new sensors are repeatable across devices and reliable over multiple cycles, but they don’t provide any experiments to show that. Yet, these features could be very valuable for the community.
- More detailed improvements are listed below.

**Quality Of The Limitations Section:**

3

**Questions For Rebuttal:**

Here is the list of additional experiments and recommendations that I believe could improve the paper:

**Tactile Sensor Related Improvements / Recommendations:**

- Could you add an experiment to showcase the transferability of the tactile readings? This could be done by repeating one of the task's results with a new set of sensors: 1) Collect demos with one set of sensors and 2) Evaluate the model trained from those demos using another set of sensors.
- Could you mention if the tactile sensors wear out over time? If so, how many cycles does it take for the sensors to wear out?
- Could you add more explanation on the calibration of the sensors, if needed? If calibration is not needed, that should also be mentioned in the paper because not requiring calibration would be valuable in itself.

**Point Cloud Generation Related Improvements / Recommendations:**

- The authors mention that they transfer the point cloud into the robot’s base frame. However, they don’t explain how they do that exactly. Do they use ArUco markers to calibrate the cameras to the environment, or do they know where the cameras are located with respect to the robot’s base? A small explanation there would be helpful.
- I am curious to see how sensitive the system is to the robot design and the sensors’ location with respect to the base. What happens if we move the location of the sensors a little bit during evaluation, for example?

**Experiments Related Improvements / Recommendations:**

- Could you include other policy rollouts of the other baselines on the website? I am especially curious how the RGB + Tactile Image baseline differs from the point cloud structure. How does the RGB + Tactile Image baseline fail compared to the PC + Tactile Points (3D-Vitac)?
- The authors have impressive single-arm tasks showcased on the website (such as turning a light bulb). I wonder if they could briefly mention them in the paper and maybe evaluate a few of the baselines on those tasks as well.

**Robotics Focus:**

4

**Summary Of Paper:**

This paper utilizes tactile sensors with resistive sensing mechanism in a bimanual gripper setting to build a 3D continuous visuo-tactile point cloud by aligning both modalities' inputs in the same space, and employs imitation learning to learn behaviors from this unified input. The novel contributions of this paper include the design of a new tactile sensor, which differs from optical and magnetic alternatives, and the use of continuous tactile values to build a 3D visuo-tactile point cloud for downstream imitation learning. This approach differs from that of Yuan et al. [27], who used binary tactile readings to build a similar unified point cloud.  The tasks evaluated by the authors are impressive, showcasing the importance of tactile feedback and the unified point cloud representation in these tasks.

**Summary Of Recommendation:**

I recommend accepting this paper, as I find the overall framework valuable to share with the community. Even though there is not significant originality in the method, a unified continuous point cloud with vision and tactile inputs has been a missing input modality for learning so far. The tactile sensor seems to be robust, the overall execution is well done, and the resulting tasks are impressive.

---

### Official Review · Reviewer_qtTt · 2024-07-22
**Review of 3D-ViTac: Learning Fine-Grained Manipulation with Visuo-Tactile Sensing**

**Originality:** 4
**Technical Quality:** 4
**Clarity Of Presentation:** 5
**Potential Impact:** 3
**Recommendation:** 3
**Confidence:** 4

**Review:**

This work introduces a 16x16 piezoresistive flat tactile array, which they install on a pair of grippers for bimanual tactile tasks, and apply visuotactile imitation learning via human teleoperation for a range of practical tasks related to food preparation.

Overall this is a strong, well-polished paper of the level for CoRL and advancing the field. An issue is there is a lot in the paper, so some key aspects are pushed into figures and/or the appendix. This is partly an issue with the expectations for CoRL.

The authors introduce a tactile array, although obviously piezoresistive skins have been around since the 1980s or earlier. It would help if the authors could emphasise their novelty here, presumably it is mainly around making the technology easy to fabricate for others. I appreciate that this is not the main contribution of the article.

I was interested to know more about the robotic system and form of the grippers, which arms are being used, the type of control available etc. I would have liked to have seen this in the main paper.

The authors combine vision and tactile into a single point cloud representation, which as I understand it is by using binary contact information at the taxels, mapped knowing the pose of the tactile pads onto the point cloud from vision. Presumably this requires accurate calibration to around the taxel size (sqrt(3)mm) - I would have liked to have known about this.

In the scenario in the paper, this makes the primary use of tactile to fill in occluded regions of the objects due to grasping, i.e. you are basically using tactile to fill in missing visual information. I'm not sure this fits so well with your motivation of the 'distinct nature of tactile and visual modalities', which I would have seen as making more use of tactile information that cannot be obtained from vision, such as contact slip, shear, force etc. Some clarification might be useful for the reader, rather than having to infer this for themselves. This also relates to your limitations in that you are only using limited tactile information, which is partly due to the nature of your tactile array being sensitive only to normal contact forces and partly due to your method using only tactile information that can be directly transformed into a visual point cloud. I would be interested to know how you would go beyond this.

**Quality Of The Limitations Section:**

2

**Questions For Rebuttal:**

See above

**Robotics Focus:**

4

**Summary Of Paper:**

This work introduces a 16x16 piezoresistive flat tactile array, which they install on a pair of grippers for bimanual tactile tasks, and apply visuotactile imitation learning via human teleoperation for a range of practical tasks related to food preparation.

**Summary Of Recommendation:**

good, well-polished paper of the level for CoRL

---

### Official Review · Reviewer_nMTy · 2024-07-24
**Nice systems paper.**

**Originality:** 3
**Technical Quality:** 3
**Clarity Of Presentation:** 3
**Potential Impact:** 3
**Recommendation:** 3
**Confidence:** 4

**Review:**

This paper presents a multi-modal, bimanual manipulation system for performing multi-modal imitation learning. The proposed sensor is based on a piezoresistive material allowing conversion of mechanical pressure to electrical signal. This sensor is attached to 3D printed soft grippers. The proposed mechanical system is used to collect multi-modal demonstration data which is used to train diffusion policy. The multi-modal policy is compared against a baseline vision-based policy to show that the tactile signals help the system to learn better policies.
Overall, the paper could be positioned as a good systems paper which clearly shows the effectiveness of using tactile signals during learning complex manipulation strategies. However, the novelty of the proposed work in limited  as it simply puts together known techniques into a system without much changes to the underlying diffusion policy. The authors show some ablations on the representation of the tactile signals which is interesting. However, it is also known that using point cloud information from tactile sensors can provide better pose resolution of objects in grasp. So, overall, the novelty of the proposed work is questionable.

**Quality Of The Limitations Section:**

1

**Questions For Rebuttal:**

Please answer the following questions:
1. What is the motivation behind the use of the proposed sensors? To make the proposal of the proposed sensors more convincing, the authors should compare the performance against the vision-based tactile sensors like gelsight. In my opinion, the proposed approach would work with other vision-based tactile sensors too. This limits the novelty of the proposed work.
2. What is the input to your diffusion training network? Is it the sequence of the visuo-tactile point cloud? or there are additional variables like grasping force?
3. It is interesting to see that some of the failure modes of the baseline (vision-only) policy is failure to grasp or grasping too tight. Im wondering how can the tactile-based policy react to such failures? Unless you allow the grasping force to the adjusted as part of the policy, it may not be possible. Can you please elaborate on this?
4. A follow up question : you show that the diffusion policy can react to failed grasp during execution. However, the vision-base policy is unreactive? like it doesn't react to the cases it fails to grasp the egg (as the arm moves, one can see the egg). please elaborate on this failure mode too.

**Robotics Focus:**

4

**Summary Of Paper:**

This paper presents a multi-modal sensing and learning system designed for bimanual manipulation. This system is equipped with new tactile sensors and the proposed system is used to perform multi-modal imitation learning on several manipulation tasks.

**Summary Of Recommendation:**

I think the paper is a systems paper as the algorithmic novelty is minimal. So, I am not sure if the paper is suitable for CoRL.

---

### Author Rebuttal · Authors · 2024-08-09

Our revised main paper and supplement are in the rebuttal file.

---

### Decision · Program_Chairs · 2024-09-04

**Decision:**

Accept

**Comment:**

The reviewers all agree that the paper provides a solid contribution and would be a good addition to the CoRL conference. The authors also did a good job in addressing clarity issues during the rebuttal and providing extensive experiments..